# High-fat diet and estrogen modulate the gut microbiota in a sex-dependent manner in mice

Linnea Hases [1,2,4], Lina Stepanauskaite [1], Madeleine Birgersson [1,2], Nele Brusselaers[3], Ina Schuppe-Koistinen[3], Amena Archer [1,2], Lars Engstrand[3] & Cecilia Williams [1,2✉]

A high-fat diet can lead to gut microbiota dysbiosis, chronic intestinal inflammation, and metabolic syndrome. Notably, resulting phenotypes, such as glucose and insulin levels, colonic crypt cell proliferation, and macrophage infiltration, exhibit sex differences, and females are less affected. This is, in part, attributed to sex hormones. To investigate if there are sex differences in the microbiota and if estrogenic ligands can attenuate high-fat diet-induced dysbiosis, we used whole-genome shotgun sequencing to characterize the impact of diet, sex, and estrogenic ligands on the microbial composition of the cecal content of mice. We here report clear host sex differences along with remarkably sex-dependent responses to high-fat diet. Females, specifically, exhibited increased abundance of *Blautia hansenii*, and its levels correlated negatively with insulin levels in both sexes. Estrogen treatment had a modest impact on the microbiota diversity but altered a few important species in males. This included *Collinsella aerofaciens F*, which we show correlated with colonic macrophage infiltration. In conclusion, male and female mice exhibit clear differences in their cecal microbial composition and in how diet and estrogens impact the composition. Further, specific microbial strains are significantly correlated with metabolic parameters.

[1] Science for Life Laboratory, Department of Protein Science, KTH Royal Institute of Technology, 17121 Solna, Sweden. [2] Department of Biosciences and Nutrition, Karolinska Institutet, 14183 Huddinge, Sweden. [3] Centre for Translational Microbiome Research, Department of Microbiology, Tumor & Cell Biology, Karolinska Institute, Science for Life Laboratory, Solna, Sweden. [4] Present address: Salk institute for biological research, La Jolla, CA 92037, USA. ✉email: cecilia.williams@scilifelab.se

The gut microbiota is important for colonic health and has been implicated in carcinogenesis. For example, germ-free mice develop more and larger tumors in a chemically induced colitis-associated colon tumor model compared to mice with intact microbiomes[1]. Further, a high-fat diet (HFD) increases the risk of developing colorectal cancer[2–4]. The colon has been identified as the first tissue to respond to a HFD, with increased pro-inflammatory signaling and colonic permeability[5]. Also, HFD is a well-known risk factor for gut microbiota dysbiosis[6]. Dysbiosis is characterized by a reduction of beneficial bacteria and an outgrowth of opportunistic pathogens. Interestingly, HFD-induced dysbiosis in mice is independent of obesity[7], which denotes the diet itself as a major modulator of the gut microbiota. A study by Liu and colleagues showed that HFD-induced dysbiosis also promoted tumor development in the $Apc^{min/+}$ mouse model, independently of obesity, by activating the monocyte chemoattractant protein-1 (MCP-1)/CC Chemokine receptor 2 (CCR2) axis[8]. This axis is involved in monocyte recruitment, which highlights the crosstalk between microbiota, inflammation, and the immune system. Multiple studies using 16S ribosomal RNA gene sequencing (16S rRNA-seq) have shown that HFD increases levels of Firmicutes and Proteobacteria and decreases Bacteroidetes (reviewed in Murphy et al.[6]), but meaningful conclusions and mechanistic insights are not possible without additional taxonomic granularity. There is not yet a consensus on how HFD affects the microbiome with this level of detail.

Results from several clinical and animal studies indicate that the sex of the host impacts the gut microbiota (reviewed in Kim et al.)[9]. The Human Microbiome Project study of healthy adults found that men had a higher relative abundance of the genus Prevotella compared to women in the same cohort, and less diversity within the genus Bacteriodetes[10]. In a 16S rRNA-seq characterization of mice, the phyla Actinobacteria and Tenericutes were shown to be more abundant in males, whereas the family Lachnospiraceae was more abundant in females[11].

It is well-known that the gut microbiota regulates estrogen metabolism and thereby impacts the available levels of both local and systemic estrogen, referred to as the estrobolome[12], but few studies have investigated whether or how estrogen impacts the microbiota composition. One study found that estrogen (17β-estradiol, E2) treatment in males reduced microbiota diversity during AOM/DSS-induced tumorigenesis and increased the relative abundance of the genus Alistipes[13]. Another study found that E2 treatment reduced evenness in ovariectomized female mice fed HFD[14].

The biological action of estrogen is mediated via three receptors, the nuclear estrogen receptors ERα (ESR1) and ERβ (ESR2), and the transmembrane G protein-coupled estrogen receptor 1 (GPER1). ERβ has been linked to anti-inflammatory and colorectal cancer-protective effects, and we have previously demonstrated that deletion of ERβ in intestinal epithelial cells enhanced tumorigenesis and cytokine signaling in both sexes in the colitis-associated AOM/DSS mouse model[15]. Using 16S rRNA-seq of stool pellets in a small study, we previously found indications that host sex and knockout of ERβ impacted the gut microbiota diversity during AOM/DSS treatment[16]. Further, treatment with an ERβ-selective agonist (DPN) attenuated HFD-induced epithelial cell proliferation and macrophage infiltration and increased the expression of circadian clock genes in the colon in a partially sex-dependent manner[17].

To test the hypothesis that estrogen signaling via ERβ-activation attenuates HFD-induced dysbiosis in a sex-dependent manner, we here applied whole-genome shotgun sequencing (WGS)[18] on cecal content from the same HFD model. We describe the complete impact that HFD and host sex have on the microbiome. To our knowledge, no studies have investigated the role of selective ERβ-activation on the gut microbiota, nor the impact of sex and estrogens using WGS, and only one study has investigated the impact of HFD on the gut microbiota in mice using WGS (stool samples)[7].

Overall, our data demonstrate significant differences in the gut microbiota composition that depend on the sex of the host. Furthermore, we denote a sex-dependent microbial response to HFD that correlates to colon cell proliferation and metabolic parameters. We identify that estrogenic ligand treatment impacted some measurements of microbiota diversity. While we found no impact by exogenous E2 on specific species in females, it significantly altered a few bacterial species in males, which correlated with F4/80$^+$ colonic macrophage infiltration, blood glucose levels, and insulin levels. The findings presented in this paper can contribute to improving knowledge of sex-dependent deleterious effects of HFD on the colon and on metabolism.

## Results

**The basic cecal microbiome is different between male and female mice.** Sixty-four cecal samples were sequenced and generated at least one million high-quality reads per sample for analysis. We first characterized the basic composition of the microbiota in C57BL/6J male and female mice (fed control diet (CD), no estrogenic ligands) using WGS and investigated sex differences. Taxonomy plots of the most abundant phyla and families are visualized and quantified in Supplementary Fig. 1. Females and males showed a relatively similar distribution, with the Bacteroidetes phylum (especially the Rikenellaceae family) being the most abundant, followed by the Firmicutes (especially the Lachnospiraceae family), Deferribacteres, Proteobacteria, and Halobacterota phyla (Supplementary Fig. 1a, c). Males presented a slightly higher ratio of Firmicutes to Bacteroidetes and a significantly higher relative abundance of Proteobacteria (Supplementary Fig. 1b), being in agreement with a previous study by Kaliannan et al. (fecal samples, 16S rRNA-seq)[19]. By plotting the principal coordinates analysis (PCoA) projection, clear sex differences were noted during CD (Fig. 1b, gray dots). We then compared differences in microbiota diversity using within-sample Shannon's diversity index that takes both the number of species and their relative abundance (alpha diversity) into account. This did not reveal a significant difference between males and females (Fig. 1a). Next, we compared the between-sample diversity (beta diversity), measured with Bray-Curtis dissimilarity. Here we found a significant difference ($P = 0.008$, 999 permutations, Fig. 1c middle panel). In order to identify which specific bacterial strains were different, we used Songbird[20], to generate differential ranking (DR) tables to identify the 10% of species that had the greatest difference between the sexes ($n = 25$). This approach identified five species that were significantly different in abundance between the sexes (adjusting for cage and diet, Fig. 1d–f). Females presented higher levels of Alistipes sp, Parabacteroides johnsonii, Bacteroides B vulgatus, and Blautia hanseniii, and males presented increased levels of an Acetatifactor species (Fig. 1f). The sex differences in the abundance of Alistipes and Parabacteroides johnsonii were confirmed using qPCR (Supplementary Fig. 2b). Since the identification of differentially abundant microbes is dependent on the method, it has been recommended to use multiple approaches for robust biological interpretation[21]. Therefore, we complemented the comparative analysis with two more methods, ANCOM-II (adjusting for both diet and cage) and DESeq2 (adjusting for diet but not cage), and focused our interpretation on species detected by at least two methods. ANCOM-II identified five species as significantly different between the sexes, and DESeq2 identified the largest difference

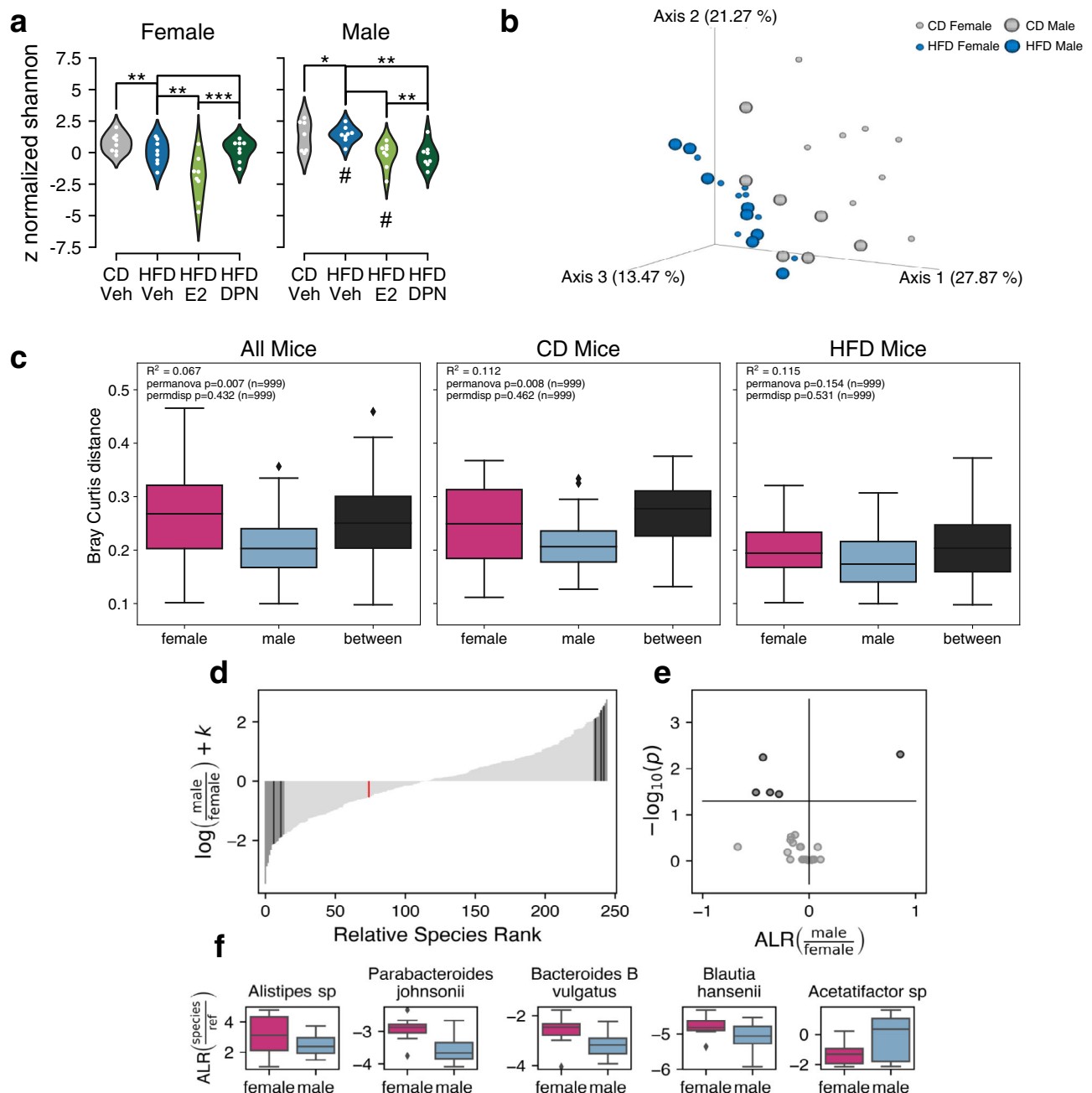

**Fig. 1 Sex differences in the microbiota composition. a** Violin plots of the alpha diversity are presented with the Shannon index (male CD: $n = 7$, all other groups $n = 8$). Student's two-tailed $t$-test, $^{*}P < 0.05$, $^{**}P < 0.01$, $^{***}P < 0.001$, $^{\#}$ indicating significant sex difference. **b** PCoA plot visualizing the beta diversity with Bray–Curtis distance in males and females on CD and HFD (male CD: $n = 7$ all other groups $n = 8$). **c** Boxplots presenting the significance of differences in beta diversity between groups, with Bray–Curtis distance. **d** Histogram plot of the species rank against the log fold change difference between males and females of the full data set ($n = 63$, adjusted for the diet/treatment interaction), along with (**e**) a volcano plot showing the significantly differentially abundant species between females and males. **f** Boxplot of the additive-log-ratio transformed taxa visualizing the significant differentially abundant species (incl. *Acetatifactor* sp002490995 and *Alistipes* sp002428825) separated on females and males (not adjusted for covariates). The boxplots are shown as median (line), interquartile range (box), and minimum to maximum data range (whisker).

(184 species), including four out of the five species identified by the DR and four out of five species identified by ANCOM-II (Supplementary Fig. 3a). The four additional species identified by ANCOM-II and DESeq2 encompassed *Lactococcus lactis* (more abundant in females) and *UBA5436* sp002427545, *CAG-776* sp000438195, and *Tranquillimonas alkanivorans* (more abundant in males). Plotting the data separated by sex and diet, showed that only one of these strains was different between the sexes under

CD (*Tranquillimonas alkanivorans*) and the remaining strains became different in response to HFD (Supplementary Fig. 3e).

Next, we compared the cecal microbiota between males and females after HFD feeding. The *Oscillospiraceae* family was significantly more abundant in males fed HFD compared to females fed HFD per taxonomy plot data (Supplementary Fig. 1c, d). However, the separation between the sexes in the PCoA plot was smaller following HFD, and no significant differences in the

beta diversity were noted (Bray-Curtis, Fig. 1b, c). Overall, the microbiome of the sexes appeared more similar to each other after HFD. However, now (during HFD), males presented a significantly higher Shannon index (Fig. 1a) and different beta diversity computed with unweighted UniFrac (Supplementary Fig. 4a, $P = 0.002$, 999 permutations) compared to females. Analysis by DR, using a sex interaction model, showed increased levels of f. Lachnospiraceae sp and decreased levels of Lactococcus lactis E in males compared to females during HFD (Supplementary Fig. 4b). The latter was also identified by ANCOM-II as described above (Supplementary Fig. 3e). Finally, when combining all mice (both diets) and investigating sample diversity between sexes, measured with Bray–Curtis dissimilarity, a significant difference remained ($P = 0.007$, 999 permutations, Fig. 1c). Thus, using WGS, we identified significant sex differences in beta diversity and for specific strains of the microbiota composition in the analyzed strain (B6) of mice.

**High-fat diet induces changes in the cecal microbiome**. As found above, the abundance of certain strains of bacteria was significantly different when the mice were fed HFD. Diet is a major modulator of the gut microbiota, and, consequently, we investigated the overall impact of HFD on the cecal microbiota content. As expected, HFD reduced both the alpha (Shannon index, Fig. 1a) and beta (Bray–Curtis, Fig. 2a left panel, sexes combined, false discovery rate (FDR)-corrected $P = 0.001$, and unweighted UniFrac distance metrics, Supplementary Fig. 4c, left panel, sexes combined, FDR-corrected $P = 0.001$, both with adjustment for cage and with 999 permutations) diversity. These data confirm previous findings that HFD has a significant impact on gut microbiota diversity. Next, we used DR to identify the 10% most extreme species ($n = 25$) associated with alterations due to HFD, after adjustment for cage and sex. Of these 25 organisms, HFD was associated with a significant loss of the relative abundance of five species, including an uncultured member of the genus Alistipes (family Rikenellaceae), two uncultured members of the genus Muribaculum, and two uncultured members of the family Muribaculaceae, along with an increase in the relative abundance of three uncultured Lachnospiraceae family species of the Sellimonas, Acetatifactor, and Faecalicatena genus (Supplementary Fig. 4d–f). An increase in Lachnospiraceae has been previously reported following HFD[22,23]. We confirmed the reduction of Alistipes and Muribaculum with qPCR in both cecal and fecal samples (Supplementary Fig. 5a, b, $P < 0.001$). qPCR analysis also supported an increase of Faecalicatena in the cecal content, but this was not significant ($P = 0.1$). Overall, our data confirmed that HFD has a significant impact on the gut microbiota, and we identified specific species that change in abundance.

**The microbial response to HFD is dependent on sex**. To investigate whether there were sex differences in the response to HFD, we used sex-stratified analyses. The overall effect of HFD on microbiota diversity was replicated in both females and males when analyzed separately (Figs. 1a and 2a, Supplementary Fig. 4c). We note that while HFD significantly reduced the Shannon index in both sexes, the reduction was stronger in females (Fig. 1a). The PCoA plot and beta diversity (Bray–Curtis) also showed a clearer separation between CD and HFD for females (Figs. 1b and 2a). There were no significant differences in dispersion in the sex-stratified analyses (Fig. 2a), making it less likely that the effect of HFD can be attributed to the within-group variation. Investigating the distribution of families, males presented a significant sex-specific increase in the Oscillospiraceae and Ruminococcaceae families (of Firmicutes phylum) upon HFD (Supplementary Fig. 1c, d). In the DR sex-stratified analysis

(adjusted for cage), we found that males presented ten species with significantly altered abundance in response to HFD, and females presented 13, of which 3 species were in common for both sexes (Fig. 2b–d). The commonly altered species included f. CAG-552, f. CAG-727 (both less abundant during HFD), and Brachyspira (more abundant during HFD, Fig. 2d). The female-specific HFD-modified species included reductions of specific species of Turicibacter, f. Muribaculaceae, Muribaculum (sp002358615), and Ruthenibacterium lactatiformans and increases of Clostridium M clostridioforme A, f Lachnospiraceae, Blautia (sp000432195), f. Ruminococcaceae, Eubacterium E, and Bifidobacterium infantis (Fig. 2d). The male-specific HFD-modified species included a reduction of f. Muribaculaceae and Agathobaculum butyriciproducens, and increases of three uncultured species of f. Oscillospiraceae, Collinsella aerofaciens F, and Ruthenibacterium (Fig. 2d). Also here, we investigated further differential abundance using ANCOM-II and DESeq2, which could validate the DR-identified HFD response in both females and males. We considered species identified by at least two methods as reliable (Supplementary Fig. 3b, Supplementary Tables 1 and 2), resulting in 26 species in males and 20 in females that were specifically altered by diet. These data thus demonstrate, to the best of our knowledge for the first time when using WGS, that there is a clear sex-dependent response to HFD in the cecal microbiome and identify numerous species that are altered specifically in either males or females.

**Estrogenic ligands have a modest impact on the microbiota composition**. As we show above, sex had a significant impact on the microbiota composition, including to its respons to a diet rich in fat, and this may, to some or all extent, be due to sex hormones. Some studies have indicated that estrogen modulates the microbiota composition, but the results are debated and have not reached a consensus. Here, we used WGS for an unbiased investigation of whether E2 or the ERβ-selective ligand DPN could modulate the cecal microbiome under HFD in either sex. The doses and treatment durations were based on earlier studies and were corroborated by their measured serum levels and physiological effects in the animals of this study[17].

As may be expected, the diet had a larger effect than the ligand treatments for all beta diversity metrics tested. Neither E2 nor DPN treatment had a major impact on the overall gut microbiome composition (compared to vehicle), as analyzed by PCoA, Bray-Curtis, and unweighted UniFrac (Fig. 3a, b, Supplementary Fig. 6a). Moreover, there were no significant bacterial species altered by the estrogenic treatments (Supplementary Fig. 6b, c, after adjusting for age and sex). However, stratification can improve the estimation of the treatment effect, especially if the effect varies between the groups. Using the sex-stratified analysis, E2 treatment in females (but not in males) and DPN treatment in males (but not in females) significantly reduced the Shannon index compared to vehicle treatment (Fig. 1a). Also, the Bray-Curtis distance indicated that the estrogenic ligands slightly compounded the effect of HFD on beta diversity (Bray–Curtis, Supplementary Fig. 3c), although the difference (between vehicle and ligand treatment) was not significant (Bray–Curtis Fig. 3a, b, unweighted UniFrac Supplementary Fig. 6a). Overall, the estrogenic ligand treatments had a larger impact in males than in the intact (higher endogenous estrogen levels) females (Fig. 3d). The taxonomic analysis of bacterial phyla and families indicated that E2 and DPN treatment both significantly reduced the relative abundance of the Oscillospiraceae family in males and that DPN treatment significantly decreased the relative abundance in males and increased the relative abundance in females of the

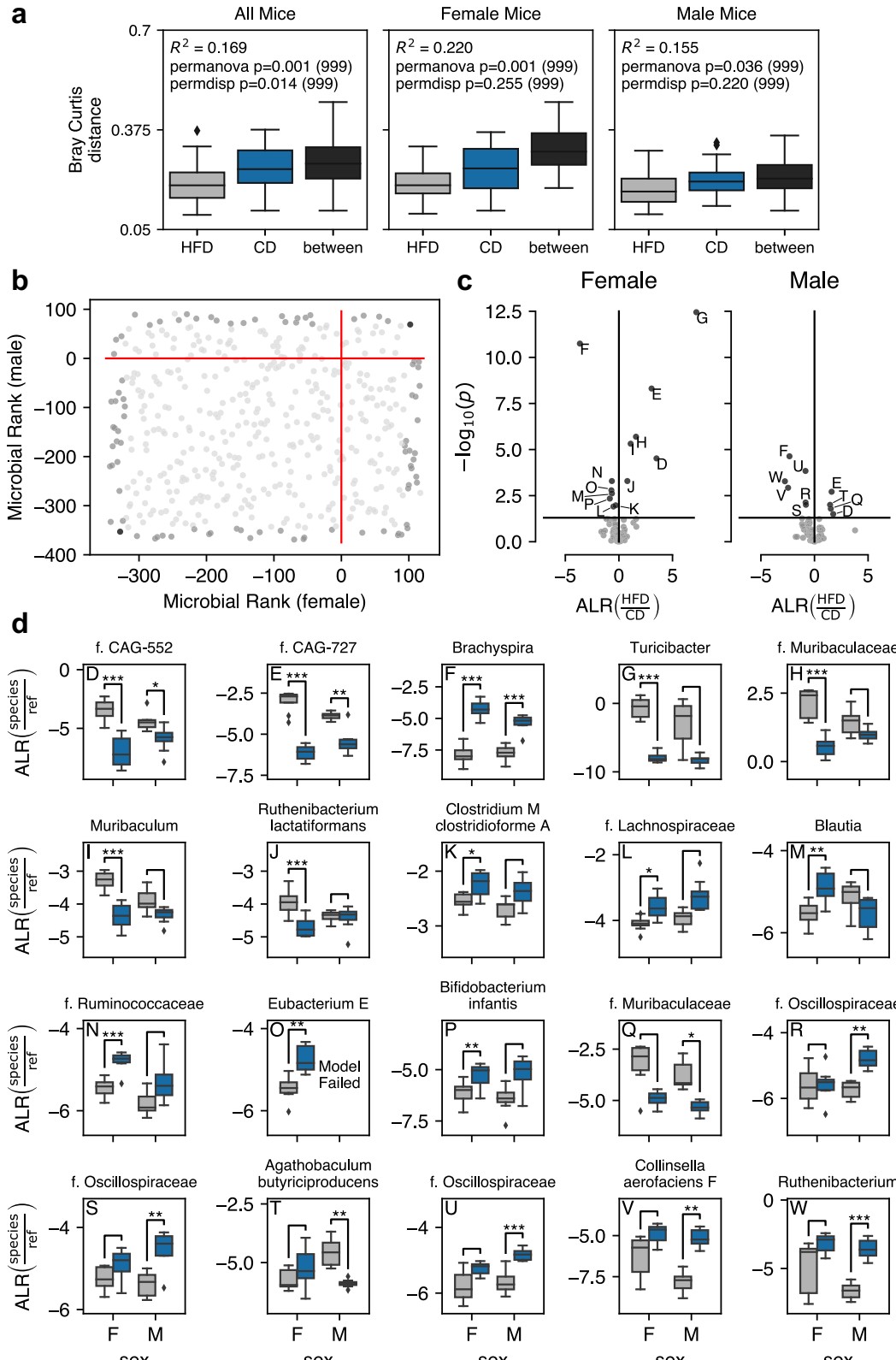

**Fig. 2 Sex-dependent microbiota responses to HFD. a** Boxplots presenting the significance of the beta diversity between groups, with Bray–Curtis distance. **b** Scatter plot of the microbial rank in males against the microbial rank in females upon HFD. The upper right corner indicates the top-ranked species in both sexes. **c** Volcano plots show the significantly altered species by HFD in females and males, respectively. **d** Boxplots of the additive-log-ratio transformed taxa showing the significantly altered species by HFD (blue) in females and males. The boxplots are shown as median (line), interquartile range (box), and minimum to maximum data range (whisker). *FDR < 0.05, **FDR < 0.01, ***FDR < 0.001 (Benjamini–Hochberg-adjusted).

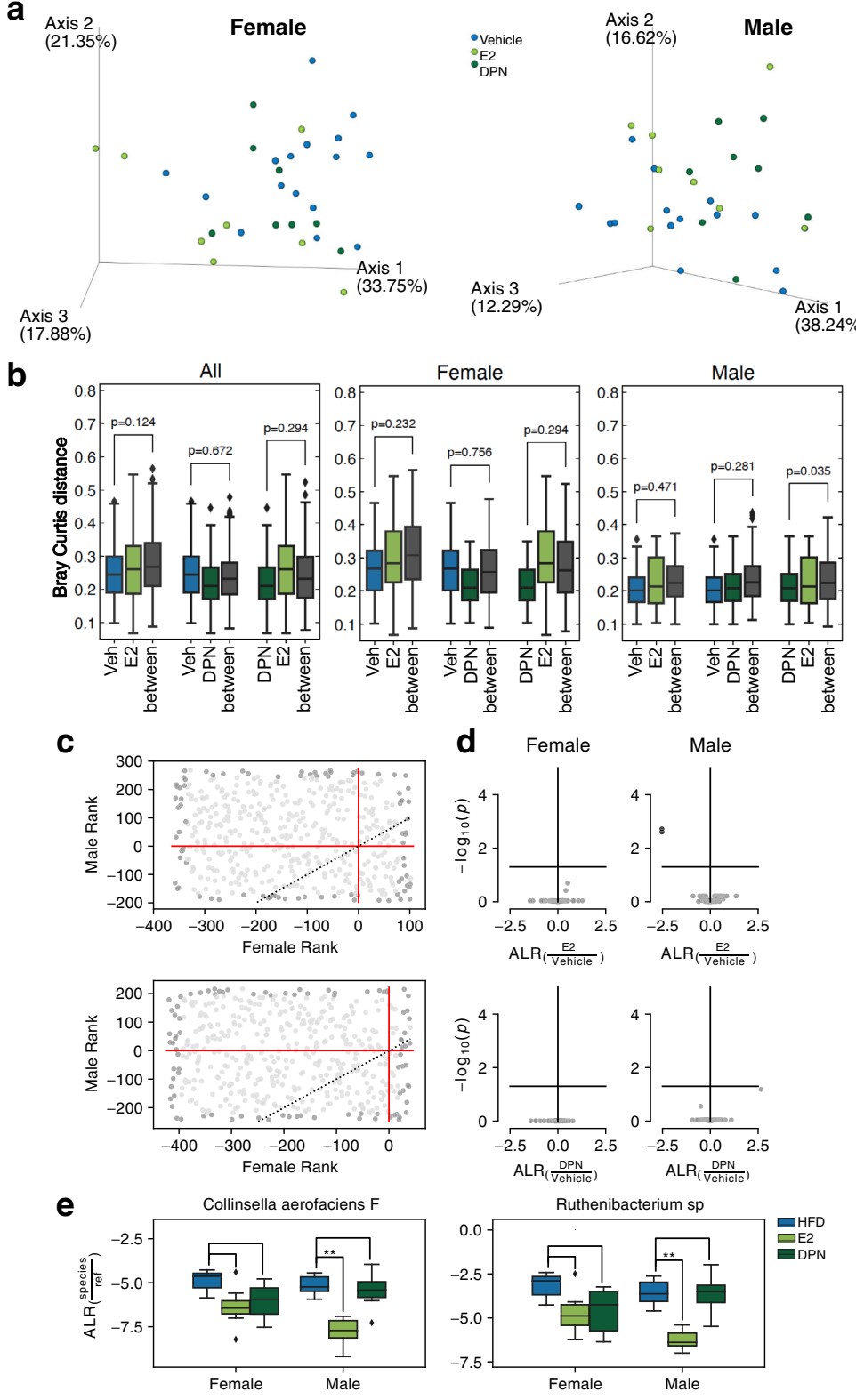

**Fig. 3 Impact of estrogen treatments on the microbiota composition. a** PCoA plot visualizing the beta diversity with Bray–Curtis distance in females ($n = 24$) and males ($n = 24$) on an HFD treated with vehicle, E2, and DPN. **b** Boxplots presenting the significance of the beta diversity between corresponding groups (all HFD), with Bray–Curtis distance. **c** Scatter plots of the microbial rank in males against the microbial rank in females upon treatment with E2 (upper panel) or DPN (lower panel). The lower right corner indicates the top-ranked species in both sexes upon E2 and DPN treatment, respectively. **d** Volcano plots show a degree of alteration of species by E2 and DPN treatment in both sexes. **e** Boxplot of the additive-log-ratio transformed taxa showing the significantly altered species by E2 and DPN treatment in both sexes ($n = 8$ per group). The boxplots are shown as median (line), interquartile range (box), and minimum to maximum data range (whisker). **FDR < 0.01 (Benjamini–Hochberg-adjusted).

*Ruminococcaceae* (Supplementary Fig. 1c, d). DR analysis identified two significantly altered species in males following E2 treatment (less abundance of *Collinsella aerofaciens F* and one uncultured species from *Ruthenibacterium*, Fig. 3c–e, Supplementary Fig. 3f). Interestingly, these two species were also significantly more abundant during HFD compared to CD in males (Fig. 2d). Applying ANCOM-II and DESeq2 analysis, we identified three additional species to be significantly altered by E2 treatment in males (by two methods, Supplementary Fig. 3c): a decreased abundance of *Anaerotruncus colihominis* and *CAG-776 sp000438195*, and increased levels of *Lactococcus sp002492185* (Supplementary Fig. 3f). *Anaerotruncus colihominis* and *CAG-776 sp000438195* were also decreased by DPN treatment (Supplementary Fig. 3f). Thus, although it has been reported that E2 can modulate the gut microbiota, we found only a relatively small impact using WGS. Estrogens thus appear to enhance the effect of HFD on both alpha and beta diversity in both sexes, but also to alter a few select species and thereby opposing the effect of HFD in a sex-dependent manner.

**Select bacterial species correlate to metabolic and phenotypic parameters**. Certain microbiota strains have been reported to aggravate or alleviate inflammatory and metabolic diseases as well as colorectal carcinogenesis. Having here identified the impact of HFD, sex, and, to some extent, estrogenic ligand treatment, on the abundance of specific microbial strains, we set out to explore whether the differentially abundant species correlated to metabolic or phenotypic parameters in the same animals. These parameters have previously been quantified and reported in Hases et al.[17]. While investigating these plausible relationships, we indeed found a significant positive correlation between three species (*Acetatifactor sp002490995, Tranquilli-monas alkanivorans*, and *CAG-776 sp000438195*) and fasting glucose levels (Fig. 4a, black dots, both sexes combined). One of these species, *Acetatifactor sp002490995*, also showed a trend of positive correlation with colonic cell proliferation (measured by Ki67 immunohistochemistry) and another, *CAG-776 sp000438195*, with insulin levels (Fig. 4b, d, black dots, sexes combined). These species were all of higher abundance in males (under HFD), and the two latter were further reduced by E2 in males (*CAG-776 sp000438195* reduced by both E2 and DPN treatment, Supplementary Fig. 3f, *Acetatifactor sp002490995* reduced by E2 in fecal samples, Supplementary Fig. 6d). Also, *Anaerotruncus colihominis* showed a positive correlation to plasma glucose levels in males (Fig. 4a, blue dots), and *Ruthenibacterium* and *Collinsella aerofaciens F* both correlated positively with F4/80$^+$ colonic macrophage infiltration (Fig. 4e, sexes combined) and the latter negatively with plasma insulin levels in males (Fig. 4b, blue dots). Interestingly, E2 treatment reduced all these species in males (Fig. 3e and Supplementary Fig. 3f).

Further, the female-abundant *Blautia hansenii* (Fig. 1f) showed a significant negative correlation with insulin levels (Fig. 4b, sexes combined, black dots) and *Lactococcus lactis* (increased by HFD, more in females, Supplementary Fig. 3e) with colonic cell proliferation (per Ki67 marker, sexes combined, Fig. 4d). In addition, *Blautia hansenii* correlated negatively with visceral adipose tissue (VAT) in relation to total fat (TF) in males (Fig. 4c, blue dots).

Our data thus demonstrates that specific strains of the male cecal microbiome correlate with higher glucose and insulin levels and increased proliferation and that treatment with E2 in males reduces the abundance of bacterial species that correlate to high insulin levels and increased colonic macrophage infiltration.

## Discussion

Diet is a well-known modulator of the gut microbiota and HFD leads to dysbiosis. In addition, sex and E2 treatment have been reported to modulate the gut microbiota but the results are controversial. Most of the studies performed in mice have used 16S rRNA-seq to profile the microbiome, which has a lower specificity in identifying specific strains, and lower sensitivity in detecting low-abundant species compared to WGS. Here we investigated the cecal microbiome of male and intact (i.e., normal hormonal levels) female mice on CD or HFD and following treatment with estrogens (E2 or DPN). In order to compare innate sex differences, none of the female mice in this study were ovariectomized.

We here corroborate that there are indeed significant sex differences in the gut microbiota composition for several species. For example, *Acetatifactor sp002490995* (family *Lachnospiraceae*), which was increased upon HFD (sexes combined), was more abundant in males compared to females. This difference may be critical since Lee et al. have shown that oral gavage of *Acetatifactor muris* aggravated DSS-induced colitis[24]. Colitis induces increased proliferation of the colonic crypt and, interestingly, *Acetatifactor sp002490995* showed a trend of positive correlation with colonic cell proliferation in our animals. The higher abundance of *Acetatifactor* in males may thus explain why males and not females presented a significant increase of colonic cell proliferation upon HFD, as previously noted[17]. Moreover, another family *Lachnospiraceae* species, *Blautia hansenii*, was more abundant in females compared to males. *Blautia* has been of particular interest because of its reported antibacterial activity and involvement in alleviating inflammatory and metabolic diseases[25]. This species has previously shown a significant negative correlation with visceral fat accumulation and fasting plasma glucose levels[26,27]. Likewise, the abundance of *Blautia* in the mucosal microbiota is reduced in patients with colorectal cancer[28]. We here demonstrate that *Blautia hansenii* negatively correlated with plasma insulin levels in both sexes. The lower levels of *Blautia* in males may thus contribute to (or be a consequence of) their increased plasma insulin levels following HFD, a sex difference previously noted[17]. Moreover, *Alistipes* were more abundant in females compared to males and have previously been shown to be increased by E2 treatment in males[13]. Our data thus present significant sex differences in the cecal microbiome, which may offer insights into the higher sensitivity of males (responding negatively) to HFD feeding compared to females.

Next, we could confirm that HFD significantly decreased the Shannon index, which is a measure of both richness and evenness (alpha diversity). Moreover, we identified that there is a sex difference in response to HFD. This is in line with our observation above, that there is a sex difference in certain bacterial strains and families at baseline (CD). For example, females fed HFD showed a sex-specific increase of three *Lachnospiraceae* species (one uncultured species, *Clostridium M clostridioforme A*, and an uncultured *Blautia* species) and one species from the *Ruminococcaeae* family. The sex-specific increase of *Lachnospiraceae* species in females supports previous findings of increased *Lachnospiraceae* upon HFD in females (per 16S rRNA-seq of fecal samples) by Qin et al.[29]. The role of *Lachnospiraceae* is controversial and it has been reported to be both beneficial and harmful to the host[30]. *Lachnospiraceae* is among the most abundant producers of short-chain fatty acids, which have several beneficial effects. On the other hand, their carbohydrate digestion (by microbes) contributes to increased energy harvest from the diet and can impact the fasting blood glucose levels. Long-term (36 weeks) HFD feeding (45%) in male mice has previously been shown to increase the abundance of *Lachnospiraceae* in fecal samples[22], and intestinal colonization with *Lachnospiraceae*

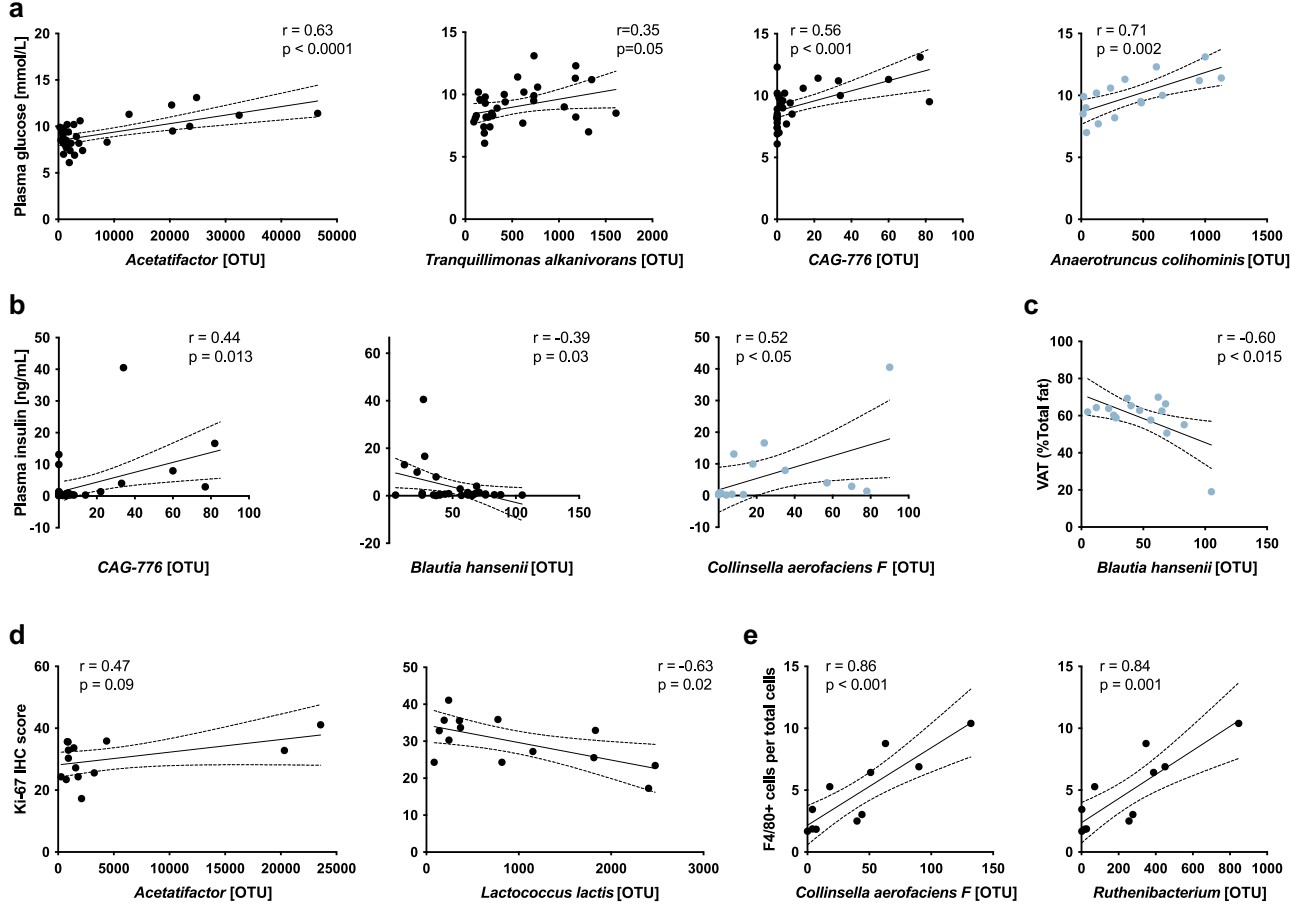

**Fig. 4 Correlation between bacterial species and metabolic or phenotypic parameters.** Bacterial species that were differentially abundant between the sexes and impacted by ligand treatment in males showed a correlation with **a** plasma glucose levels $n = 32$ sexes combined, $n = 16$ for males only in blue), **b** insulin levels ($n = 31$ sexes combined, $n = 15$ for males only in blue), **c** visceral adipose tissue (VAT) in relation to total fat (n = 16, males only), **d** with Ki67 immunohistochemistry (IHC) proliferation score ($n = 14$, sexes combined), and **e** with F4/80$^+$ macrophage colonic infiltration ($n = 11$ sexes combined). Metabolic and phenotypic parameters were previously analyzed and reported[17]. The best fit line was fitted with 95% confidence intervals, $r$ Pearson correlation coefficient, and $P$ value from a two-tailed Student's $t$-test.

contributed to the development of diabetes in obese mice[31]. Moreover, we have identified higher levels of *Lachnospiraceae* in intestinal-specific ERβ knockout mice, who also demonstrated increased tumorigenesis during AOM/DSS treatment[16]. In our current study of the cecal microbiome, not only did the *Lachnospiraceae* family appear to increase following HFD overall (Supplementary Fig. 1c, d), but in addition to the female-specific changes noted above, we observed particularly strong general increases in the species *Sellimona*, *Acetatifactor*, and *Faecalicatena* (Supplementary Figs. 4f and 6d), along with an uncultured species of the *Lachnospiraceae* family (Fig. 2d).

Moreover, females presented sex-specific decreases of *Rikenellaceae* and *Ruthenibacterium lactatiformans* (family *Ruminococcaeae*) upon HFD. *Ruthenibacterium lactatiformans* is a lactate producer, and lactate treatment has been shown to prevent chemically induced colitis in mice[32]. Males, on the other hand, presented sex-specific increases of three *Oscillospiraceae* species, *Collinsella aerofaciens F*, and a species of *Ruthenbacterium* (family *Ruminococcaea*). Males and females thus exhibited differences in species of *Ruminococcaeae*, which is the main family that converts primary bile acids to secondary bile acids. The secondary bile acid, deoxycholic acid (DCA), induces DNA damage and proliferation in Lgr5$^+$ cancer stem cells and promotes adenoma-to-adenocarcinoma progression[33]. Furthermore, a deregulated bile acid-gut microbiota axis has been reported to

affect obesity susceptibility[34]. Our results, in part, differentiate from a previous HFD study (12 weeks, 16S rRNA-seq, cecal content, males), which found increased abundance of *Rikenellaceae* (we found female-specific decrease) and *Alistipes* (we found decreased levels of specific species) and decreased abundance of *Ruminococcaceae* (we found a sex-specific increase of this family in males)[35].

Our study is the first to present a sex-dependent response to HFD using WGS. One previous study has reported sex-specific alterations in the gut microbiota (45% HFD for 16 weeks) using 16S rRNA-seq of stool samples[36]. In that study, Peng and colleagues noted a sex-specific increase of *Blautia* in females, which we corroborate. While these two studies investigate different microbiomes (fecal and cecal, respectively), both determine that sex has a large impact on the microbiota. Peng and colleagues report this effect as larger than the effect by HFD, while we found that HFD in females mediated the strongest impacts on the beta-diversity and that sex had the second impact in the absence of HFD (as inferred from PCoA in Fig. 1b and Bray–Curtis distance in Figs. 1c and 2a).

The strengths of our study include that we used WGS and can identify specific species, and importantly, that we used large groups and could control for cage effects. Mice housed at the same facility but in different cages are known to differ in microbiota composition, which can account for up to 30% of the

compositional variance[37,38]. Such cage effect is exemplified also in our study by the variance in diversity (unweighted UniFrac, weighted UniFrac, and Bray–Curtis dissimilarity) being larger between cages than within the cages (Supplementary Fig. 2a). To account for this, we had at least two cages of animals for each treatment and sex combination, although mice were separated by sex. We also included diversity in our models, either by adjusting (beta) or using a mixed-effect model that treated the cage as a random intercept. This all contributes to the rigor of our study. Worth mentioning is that E2, DPN, and vehicle-treated mice were housed in different cages, but the impact of treatment was notable only in males (Fig. 3). The estrogens did not separate the overall composition (PCoA), impact the beta diversity, or the abundance of specific species in the females. This is in accordance with the females maintaining endogenous estrogen levels since they were intact (i.e., not ovariectomized). This supports that we could successfully control cage effects.

We also investigated whether estrogens contribute to the noted sex differences in the cecal microbiome. Previously, E2 treatment has been shown to reduce the alpha diversity (16S rRNA-seq) in males during AOM/DSS-induced tumorigenesis[13], and in ovariectomized female mice fed HFD[14]. We could confirm that E2 treatment in females on an HFD significantly reduced the alpha diversity (Shannon index). E2 treatment in males showed a non-significant trend of reducing alpha diversity, whereas, interestingly, the ERβ-selective agonist DPN reduced the alpha diversity significantly. Furthermore, the ligand treatments show a larger Bray–Curtis distance to CD compared to vehicle treatment. These results demonstrate that the ligands compounded the effect of HFD in this respect. However, we also found that E2 treatment could oppose the HFD-induced abundance of certain species, including *Collinsella aerofaciens F* and *Ruthenibacterium* sp002315015 in males. The fact that E2 opposed the HFD-increased abundance of *Collinsella aerofaciens F* may have implications for the insulin levels since both we (in male mice) and Gomez-Arango and colleagues (in overweight and obese pregnant women[39]) show a positive correlation between *Collinsella* and insulin levels. This connection between *Collinsella aerofaciens F*, insulin levels, and estrogens may thus help explain the significant increase of insulin levels in the males following HFD, as previously reported[17]. *Collinsella aerofaciens* is also enriched in irritable bowel syndrome patients[40], which aligns with its correlation to F4/80$^+$ colonic macrophage infiltration as observed in our study. Moreover, we found that E2 treatment in males reduced the abundance of *Anaerotruncus colihominis* and *CAG-776* sp000438195, which both correlated with plasma glucose levels (*Anaerotruncus colihominis* specifically in males), and *CAG-776* sp000438195 also with plasma insulin levels. Our results thus demonstrate that E2 treatment in males could oppose the HFD-induced effects of specific species that showed a positive correlation with plasma insulin, glucose, and F4/80$^+$ colonic macrophage infiltration. We found no significant altered species upon estrogenic treatment in females. Since the females were intact with already higher levels of endogenous estrogens (previously measured and reported in[17]), this may be the reason for the lack of estrogenic effects to be detected as significant. We thus find that estrogen indeed contributes to aspects of the sex difference in the microbiota. Although its effect does not appear to be the major reason for the large microbial sex difference portrayed (estrogen did, for example, not have a significant impact on beta diversity), it did affect key strains which have been linked to both metabolic and inflammatory phenotypes. However, further probiotic studies are needed to determine the exact role of the different species during HFD-induced colon inflammation.

Estrogen activates its three receptors throughout the body and can have both local and systemic effects, in part depending on where the receptors are expressed. ERβ is the predominant estrogen receptor in the intestines, where it has local anti-inflammatory effects[15]. ERα, on the other hand, is expressed at higher levels in multiple other tissues[41] and may have larger systemic effects. We do not know, at the molecular level, how the noted effects on the microbiome are mediated. From our study, we can conclude that DPN (activates ERβ) had a significant impact on alpha diversity in males, along with an impact on a single bacterial species, but the larger microbiota composition effects were noted following E2 (activates both ERα, ERβ, and GPER1) treatment. This suggests that the systemic (mostly ERα or GPER1 mediated, based on receptor expression) estrogenic effects may be the larger contributo, to the microbiota diversity.

A limitation of our study is that we did not take additional precautions to further limit differences in microbiota between cages, such as mixing bedding between cages or co-housing prior to the start of the study. However, as discussed above, adjustment for the cage effect was applied statistically. The outcome suggests that this could effectively remove cage effects since females treated by vehicle (two cages) compared to females treated with E2 (three cages), did not exhibit significant differences. Although further precautions certainly could have strengthened the rigor, it appears that the statistical adjustment has been efficient. Further, estrogen has been reported to impact the feeding behavior[42], which in turn could impact the microbiome. However, as food intake was not successfully monitored in this study (as reported previously[17]), we cannot exclude the possibility that some of the modest effects of estrogen could be through decreased diet consumption. Another potential weakness of our study is that we did not investigate the mucus microbiome. There are physiological variations in chemical and nutrient gradients and differences in host immunity along the lengths of the small and large intestines. These factors are known to impact the microbiome, which may thus be different in different parts of the intestinal system. The colonic mucus microbiome has been shown to more closely correlate with disease severity compared to alterations in the fecal and cecal microbiome in mice with chemically induced colitis. Hence, sampling of the colonic mucosa is important, and the cecal and fecal samples are not always a good proxy for the microbiome[43].

The gut microbiota has been implicated in the carcinogenesis of the colon. Also, lifestyle factors, including a diet rich in fat, have been reported to increase colorectal cancer rates among young adults[44], whereas hormones including oral contraception have been reported to reduce incidences[45]. It is thus important to study how HFD and hormones impact the microbiome, including in premenopausal women. Our study layout has allowed for a clear interpretation that HFD indeed strongly impact the microbiome of both male and females, in a sex-dependent manner, whereas estrogens have only modest effects. The estrogenic effects on the microbiome are especially small effect in intact females (equivalent to premenopausal women). Thus, the preventive effect of estrogens may not include changes in the microbiome to a great extent. However, specific strains regulated by estrogens (particularly in males) did show an especially strong correlation to important physiological parameters, such as inflammation (macrophage infiltration) and proliferation. Therefore, it cannot be excluded that estrogens impact key bacterial strains that may, in turn impact carcinogenesis.

In conclusion, our data demonstrate that there are significant sex differences in the microbiota composition at baseline (during CD) along with sex-dependent responses to HFD. We demonstrate that estrogens partly contribute to these differences and E2 treatment in males could oppose specific HFD-induced species that positively correlated to metabolic and inflammatory parameters. Our findings provide insights into the sex-dependent

deleterious effects of HFD and highlight the importance of considering sex in study designs.

## Methods

**Animal model**. Five- to six-week-old male ($n = 32$) and female ($n = 32$), C57BL/6J mice were obtained from in-house breeding. Littermate animals were group-housed under a specific pathogen-free (SPF) controlled environment at 20 °C with a 12-h light-dark cycle in individually ventilated cages. Bedding and cages were sterilized by autoclaving, the diets were irradiated, and the water was filtered and implemented with sodium hypochlorite. The animals were fed an HFD (D12492, 60% kcal fat, Research Diet) or a CD corresponding to a matched low-fat diet (D12450J, 10% kcal fat, Research Diet) for 13 weeks, and water was provided *ad libitum*. After 10 weeks of diet, the mice were injected intraperitoneal every other day for a total of 9 injections with different estrogenic ligands: 0.05 (females) or 0.5 mg/kg (males) body weight for 17β-estradiol (E2, Sigma-Aldrich), 5 mg/kg body weight for 2,3-*bis*(4-Hydroxyphenyl)-propionitrile (DPN, Tocris), or vehicle. The ligands were prepared in a solution of 40% PEG400, 5% DMSO, and 55% water. Each treatment group included 8 mice per sex. The mice were sacrificed at the end of the experiment (13 weeks, the final average body weights of corresponding groups are provided in Supplementary Table 3). The study complied with relevant ethical regulations for animal testing and research and was approved by the local ethical committee of the Swedish National Board of Animal Research (N230/15), and the animal experiment, including metabolic and phenotypic parameters, has been reported previously[17].

**DNA extraction, library preparation, and sequencing**. Half of the cecal content was used for DNA extraction. For the fecal samples, half of the content for each fecal pellet was pooled together to accommodate for possible differences in microbiota diversity between fecal pellets. DNA was extracted using the QIAmp DNA stool mini kit (Qiagen, Sweden) according to the manufacturer's recommendations. DNA concentration and purity were measured using Nanodrop 2000 spectrophotometer (Thermo Scientific). Eight cecal samples of each sex and condition ($n = 64$) were prepared for sequencing. DNA sequencing was performed using a $2 × 100$ strategy on the MGI G400 platform using the G400 sequencing kit according to the manufacturer's instructions. The resulting FASTQ sequences files and corresponding metadata are uploaded to the public European Nucleotide Archive (ENA) database (accession number PRJEB52269).

**Sequence annotation**. Samples were processed using the StaG pipeline [10.5281/zenodo.3673735]. Briefly, demultiplexed reads were quality trimmed using fastp; host removal was performed using kraken2[46,47]. Reads were annotated using kraken2 with the 89th release genome taxonomy database (GTBD), an exclusively bacterial database[47,48]. The table was collapsed to the genome level annotation for analysis (S1 level), where the taxonomic string was constructed by assembling previously listed levels. All samples had more than one million high-quality reads annotated; we did not exclude any samples due to sequencing depth. A single mouse from a cage from CD in males was selected for sequencing (although the animal was originally co-housed). This animal was excluded from further analysis to allow modeling to adjust for the cage effects (Supplementary Fig. 2). Mice from two to three cages per condition were used for the sequencing to minimize the effect of the cage.

**Diversity measurements**. Samples were rarified to 1,100,000 sequences per sample. Alpha diversity was initially calculated on the genome and species level. We evaluated observed genomes/species, using Shannon diversity, which considers both richness and evenness[49]. Metrics were calculated using the q2-diversity plugin in QIIME 2 (v. 2020.2)[50]. Alpha diversity measurements were z-normalized against vehicle-treated females fed HFD. Beta (between-sample) diversity was evaluated using unweighted UniFrac and Bray–Curtis distances on the rarified table using the q2-diversity in QIIME 2[51,52]. Bray–Curtis distance accounts for the relative abundance of taxa but does not take into account their relatedness, and unweighted UniFrac considers shared phylogenetic history along with presence and absence.

**Statistics and reproducibility**. Analyses were conducted comparing the microbiome of the two dietary groups (CD vs. HFD) within the vehicle-treated mice, as well as comparing treatment (vehicle, E2, and DPN) among mice fed an HFD. A total of 64 mice were analyzed, and one-time point (one cecal microbiome per mouse, taken at sacrifice) was divided into groups of 8 for each sex and condition. One mouse (male, CD) was removed due to a lack of cage control for this mouse. In both conditions, we fit three models of linear mixed-effects models: the depth and sex-adjusted covariate; the depth-adjusted interaction with sex; and models stratified by sex. These models treated the cage as a random intercept. Alpha diversity was compared using a linear mixed-effects model with a cage as a random intercept. Tests were conducted using statsmodels (v. 0.11.1) in python 3.6.7 [Seabold]. A $P$ value of 0.05 was considered significant. Plots were made using Seaborn (v 0.1.0) and Matplotlib (v. 3.1.3)[53,54]. We compared beta diversity using a multivariate permanova with 9999 permutations with the R vegan library (v. 2.5.6)[55]. We tested the assumption of within-group homogeneity using the permdisp test in scikit-bio comparing the centroid with 999 permutations (www.scikit-bio.org)[56]. Principle coordinate projections were calculated using q2-diversity; PCoAs were visualized using q2-emperor[57]. Boxplots of within and between-group dispersion were made with seaborn and matplotlib. A $P$ value of 0.05 was considered significant. In all cases, measurements were taken from distinct samples.

**Differential abundance testing**. A two-step differential abundance test was used to look for individual organisms associated with differences. Species present with a relative abundance of greater than 1e−5 in at least 12 taxa were retained for testing; the remaining counts were combined into a reference bin. We applied DR using Songbird[20], and genome-level relative abundance was modeled using a negative binomial. We started with a naïve assumption that there was between a 0 and 5-fold change in a given organism and used 2000 iterations to improve this fit. Modeling was performed in pystan (Stan development team 2018, v. 2.17.1.0)[58]. Taxa were ranked based on their relative effect size after fitting. The whole dataset, adjusted for diet/treatment interaction, was used for ranking. We selected the top 10% most differentially abundant taxa for a given set of covariates for further testing. The abundant-feature table was transformed using an additive-log ratio (ALR) with the collapsed remaining non-abundant taxa as a reference group, and the ALR-fit taxa were tested using a linear mixed-effects model with statsmodels. A Benjamini–Hochberg-adjusted $P$ value (FDR) of 0.05 was considered significantly different. DESeq2 (v1.30.1)[59] and adopted ANCOM-II (v2.1)[60] methods were also used to test differential abundance between two groups of interest. Both used operational taxonomic units (OTU) counts as input, and the Benjamini–Hochberg procedure to estimate FDR. For DESeq2, taxa were considered significantly differentially abundant if FDR < 0.05 and |log2FC| > 0.4. For ANCOM-II, the main significance was also calculated by Wilcoxon, which returned $W$ values (the number of times the null hypothesis, of no differential abundance, was rejected) for significance determination. Before the ANCOM-II analysis, data was manually filtered out to remove taxa with less than 100 OTU counts between all samples. In the ANCOM-II analysis, the cage was used as a random intercept. For the sex differential analysis, the diet was adjusted. Taxa were considered as significantly differentially expressed if FDR < 0.05 and $W > 0.7$.

**Taxonomy plots**. OTU values for the phyla and families were summed up, and the percentages were calculated based on the total sequenced OTUs. Phyla and families that constitute more than one percent of the total OTUs were plotted in the taxonomy plot, and the rare ones (<1%) were grouped together and plotted as others. One-way ANOVA with an uncorrected Fisher's least significant difference (LSD) test was used. A $P$ value of less than 0.05 was considered significant.

**qPCR analysis**. qPCR was used to validate the WGS data and to compare cecal content with fecal samples. Primers for *Faecalicatena* and *Muribaculum* genera, *Alistipes* sp002428825, *Parabacteroides johnsonii, Acetatifactor* sp002490995, and total bacteria were designed (primer sequences can be found in Supplementary Table 4). Primers for the bacterial DNA were designed using NCBI Primer-BLAST and RDP software. Representative taxonomic rRNA sequences were found using the NCBI taxonomy browser. For species, that did not have representative rRNA sequences, part of the recorded contig sequence was used. The specificity of the primers was validated by alignment to the ribosomal database project (RDPII) with the probe match tool. The number of targeted species was noted in cases where the primer was aimed to cover the family or genus of bacteria, and the relative coverage percentage was calculated. Specific DNA was targeted, amplified, and detected using qPCR. Five to 10 nanogram of DNA was used to perform qPCR in a total volume of 10 µl in the CFX96 Touch System (Bio-Rad), with iTaq Universal SYBR Green supermix (Bio-Rad) as recommended by the supplier. Samples were run in duplicate, and the relative expression was calculated as the mean per group using the ΔΔCt method, normalized to the mean of total bacteria (total genomic 16S rRNA). Dissociation curve analysis was done to ensure the amplification of a single amplicon. Testing of significance between the two groups was performed using the two-tailed Student's *t*-test, and a $P$ value of less than 0.05 was considered significant. In all cases, figures illustrate measurements taken from distinct samples. Cage control or batch adjustments were not done for qPCR analysis.

**Correlation with metabolic and phenotypic parameters**. Specific bacterial strains that were identified to be significantly differentially abundant were correlated to the metabolic and phenotypic parameters of the same mice. The parameters were analyzed and reported as previously described[17]. In short, blood glucose levels were measured following fasting (6 h) using a OneTouch Ultra glucometer (AccuChek Sensor, Roche Diagnostics). For insulin measurements, the ELISA kit (EZRMI-13K) was used on plasma collected from blood taken at sacrifice by cardiac puncture. TF was calculated as the sum of all fat depots (dissected from the abdominal and posterior subcutaneous regions and weighted), including gonadal fat, retroperitoneal fat, omental fat, and inguinal/gluteal fat depots, VAT comprised the sum of gonadal fat, retroperitoneal fat, and omental fat, whereas subcutaneous fat (SAT) included the inguinal/gluteal fat depots. IHC analyses were performed using primary antibodies for F4/80$^+$ (1:100, cat# MCA497R, lot# 1608, RRID: AB_323279, Bio-Rad), and Ki67 (1:100, SP6, cat# MA5-14520, lot#

TC2542944A, RRID AB_10979488, Invitrogen) on formalin-fixed paraffin-embedded colons rolled into a Swiss roll.

**Reporting summary**. Further information on research design is available in the Nature Research Reporting Summary linked to this article.

## Data availability

The data that support the findings of this study are openly available in the European Nucleotide Archive (ENA) database at https://www.ebi.ac.uk/ena/, reference number [PRJEB52269]. The QIIME2 tables underlying the analyses are available in Supplementary Data 1 and 2. All other data are available from the corresponding author upon reasonable request.

## Code availability

The analysis methods and software used in this article are all open sources, and no new methods or algorithms were generated.

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

## Acknowledgements

We would like to thank Dr. Marion Korach-André, Christina Savva, and Rajitha Indukuri (Karolinska Institute) for their assistance with the animal experiment. Sequencing and data analysis was performed by the Centre for Translational Microbiome Research at the Karolinska Institute, and we are thankful for assistance with the data analysis (Dr. Justine Debelius) and general input (Dr. Fredrik Boulund). The computations and data handling were enabled by resources in projects SNIC 2017/7-414 and SNIC 2021/22-333 provided by the Swedish National Infrastructure for Computing (SNIC) at UPPMAX, partially funded by the Swedish Research Council (grant agreement no. 2018-05973). This work was supported by the Swedish Cancer Society (21 1632 Pj), Swedish Research Council (2017-01658 and 2022-00901), Stockholm County Council (2017-0578), and Karolinska Institutet Ph.D. student support (KID 2-5586/2017 and 2021-00501).

## Author contributions

L.H., L.S., M.B., and A.A. performed the experiments; L.H. and L.S. did data analyses; C.W. conceived the research and supervised; C.W., A.A., and I.S.K. designed the study. L.H., L.S., A.A., C.W., I.S.K., N.B., and L.E. interpreted the data. L.H. and C.W. wrote the paper. All authors read, commented, and/or edited the paper.

## Funding

## Competing interests

The Centre for Translational Microbiome Research (CTMR) and associated authors (N.B., I.S.K., and L.E.) are partly funded by Ferring Pharmaceuticals. The remaining authors declare no competing interests.
