## [Peer Review File · Communications Biology]

Reviewers' comments:

Reviewer #1 (Remarks to the Author):

The manuscript reported the effects of sex differences in cecal microbiota and found estrogenic ligands can attenuate HFD-induced dysbiosis. The manuscript was well written, I only have some minor concerns.

Minor comments:

1. Experimental results should be in the past tense. Citing other people's research should also be in the past tense.
2. Line 26, "...we note..." should be changed to "...we noted...".
3. Line 31-33, "microbial sex differences" can confuse others. There are host sex differences in response to HFD and estrogens by microorganisms.
4. The logic and rigour of the introduction are not strict enough. e.g. line 41-42, What the author meant was that the colon increased intestinal permeability.
5. In the introduction section, the author introduced the relationship between BMI and CRC, but this paper has nothing to do with CRC. The author should focus on intestinal inflammation caused by HFD.
6. Line 104, "can contribute to improved knowledge of" should be changed to "can contribute to improving knowledge of".
7. In the materials and methods section, how many mice in each group, and how many are male and female? specific information should be displayed here, such as body weight, daily food intake, and sampling time points.
8. List the full name of the abbreviation when they first appear, e.g. Line 473, CD? Line 476 i.p.?
9. Figure 4 abscissa legend needs to be listed.
10. Format of references, e.g. Author name of ref 35? Periodical name and year of ref 62? Missing page number of ref 12.

Reviewer #2 (Remarks to the Author):

The manuscript by Hases et al explored the effect of a high fat diet upon metabolic, immune, and intestinal mucosal proliferation in male and female mice. The authors explored these variables with the premise that sex dependent differences in disease risk, such as colorectal cancers, are observed in humans. The authors determined sex dependent differences in microbiota community structure under control diet, where few changes were observed at the phylum level, and some species level differences were observed between sexes at baseline. Upon transitioning to a high fat diet, sex dependent differences were observed in alpha diversity, beta diversity and numerous species. Finally, using estrogenic ligands, the authors determined greater community changes in males, followed by interesting correlations between various microbiota community members and insulin levels, epithelial cell proliferation, and macrophage infiltration in the colon. While these data are very descriptive, the use of WGS and the size of the experimental groups presents high quality data to the reader. Appropriate computation and statistics were performed to verify species specific differences. The authors conclude changes in microbiota community membership may be driven by sex dependent variables, and responsible for metabolic parameters in response to diets. The manuscript is well written although the discussion is quite long. Limitations are appropriately included and raw sequencing data are publically available.

I have no major comments for the authors, but a series of minor comments.

One of my biggest challenges with this paper is interpreting the figures presented. For instance, line 131 states different approaches were used to determine sex dependent differences, shown in figure one D-F. Since figure 1 also includes control and high-fat diets for both sexes, can the authors add a few clarifying comments whether figures 1 D-F is on high-fat diet or combined?

In figure 2C it is almost impossible to read some of the labels in the volcano plot.

The data presented in figure 4 show species level correlations between various metabolic and phenotypes. However, unless I am missing it, I do not see any details or description of the methods used for plasma glucose and insulin levels, or immunohistochemistry of Ki67 or at F4/80 staining in the colon tissues in the methods section. This should be included.

In the same figure 4, best fit or confidence intervals might benefit the reader in addition to the R and P values provided.

On page 20, it is unclear if littermates were used in all studies.

Reviewer #3 (Remarks to the Author):

The presented studies were designed to determine if sex differences influences gastrointestinal microbiota populations and whether estrogenic ligands can attenuate high fat diet (HFD)-induced dysbiosis. Whole-genome shotgun sequencing (WGS) was used to define microbial composition of cecal samples in an effort to characterize the impact of HFD, gender, and estrogenic ligands and the interplay between the variables. Sex differences were observed at baseline and HFD induced changes in microbiota populations, both of which support previously published studies. Estrogen ligands had some effects particularly in male mice and some novel data is presented. However, there are questions and concerns related to the study design and data analysis that should be addressed to enhance the manuscript.

Major Concerns:

1. One primary concern is the lack of description for the methodology for how procedures were followed to minimize differences in the microbiota of test animals that were not relevant to the hypotheses proposed. The authors discuss how efforts were made to account for cage effects statistically. However, best practice is to employ methodology (i.e. mixing bedding between cages, co-housing prior to start of the study, autoclaving bedding and water, etc...) that minimizes such differences. What such methodology was used should be discussed.
2. The biological relevance of the estrogen ligand exposure is also a concern. It is not clear what exposing an intact female or male mouse to estradiol models for. In both cases, other hormones will be elevated which is likely to confound the data and make it difficult to interpret. This would apply to a lesser extent to treating with DPN in the same animals. More discussion should be provided as to why this experimental design was used as opposed to using the ligands in a low estrogen environment (i.e. ovariectomized mice).
3. Because estrogen is thought to influence it in rodents, was food intake measured? Changes in food intake might influence the microbial composition of the gut.

Minor concerns:

1. There are a few minor typographical errors that should be addressed in editing.

Point-by-point response to Reviewers' comments:

Reviewer #1: The manuscript reported the effects of sex differences in cecal microbiota and found estrogenic ligands can attenuate HFD-induced dysbiosis. The manuscript was well written, I only have some minor concerns.

Minor comments:

1. Experimental results should be in the past tense. Citing other people's research should also be in the past tense.

Authors: We completely agree and have edited this throughout the manuscript.

2. Line 26, "...we note..." should be changed to "...we noted...".

Authors: Agree and edited.

3. Line 31-33, "microbial sex differences" can confuse others. There are host sex differences in response to HFD and estrogens by microorganisms.

Authors: We have clarified this throughout.

4. The logic and rigour of the introduction are not strict enough. e.g. line 41-42, What the author meant was that the colon increased intestinal permeability.

Authors: We have clarified this.

5. In the introduction section, the author introduced the relationship between BMI and CRC, but this paper has nothing to do with CRC. The author should focus on intestinal inflammation caused by HFD.

Authors: We have removed this section.

6. Line 104, "can contribute to improved knowledge of" should be changed to "can contribute to improving knowledge of".

Authors: This has been edited.

7. In the materials and methods section, how many mice in each group, and how many are male and female? specific information should be displayed here, such as body weight, daily food intake, and sampling time points.

Authors: We apologize for omitting this essential information. The number of mice (32 male, 32 female), sample point (13 weeks from start of diet) and body weights (Supp table S3) has been added (M&M, p. 21). Regarding daily food intake, see comment 3 to Reviewer 3 below.

8. List the full name of the abbreviation when they first appear, e.g. Line 473, CD? Line 476 i.p.?

Authors: This has been adjusted (CD for Control Diet, i.p. has been replaced with intraperitoneal injection).

9. Figure 4 abscissa legend needs to be listed.

Authors: Abscissa legend OTU (for operational taxonomic units) has been added, and an updated Figure 4 is provided below.

10. Format of references, e.g. Author name of ref 35? Periodical name and year of ref 62? Missing page number of ref 12.

Authors: This has been adjusted (reference 12 is removed, per comment #5).

Reviewer #2: The manuscript by Hases et al explored the effect of a high fat diet upon metabolic, immune, and intestinal mucosal proliferation in male and female mice. The authors explored these variables with the premise that sex dependent differences in disease risk, such as colorectal cancers, are observed in humans. The authors determined sex dependent differences in microbiota community

structure under control diet, where few changes were observed at the phylum level, and some species level differences were observed between sexes at baseline. Upon transitioning to a high fat diet, sex dependent differences were observed in alpha diversity, beta diversity and numerous species. Finally, using estrogenic ligands, the authors determined greater community changes in males, followed by interesting correlations between various microbiota community members and insulin levels, epithelial cell proliferation, and macrophage infiltration in the colon. While these data are very descriptive, the use of WGS and the size of the experimental groups presents high quality data to the reader. Appropriate computation and statistics were performed to verify species specific differences. The authors conclude changes in microbiota community membership may be driven by sex dependent variables, and responsible for metabolic parameters in response to diets. The manuscript is well written although the discussion is quite long. Limitations are appropriately included and raw sequencing data are publically available.

I have no major comments for the authors, but a series of minor comments:

1. One of my biggest challenges with this paper is interpreting the figures presented. For instance, line 131 states different approaches were used to determine sex dependent differences, shown in figure one D-F. Since figure 1 also includes control and high-fat diets for both sexes, can the authors add a few clarifying comments whether figures 1 D-F is on high-fat diet or combined?

Authors: We have now attempted to clarify this, included in figure legends and methods section (p. 22 and 24). Figure 1D-F visualizes the full dataset (both diets and all treatments). The data in two of these subpanels, D-E, are adjusted for the diet/treatment interaction and include all four combinations (CD vehicle, HFD vehicle, HFD E2, and HFD DPN). Figure 1F visualizes the pooled data separated by sexes and not adjusted for any covariates. (Separation on diet and treatments are further shown in Figures 2-3 and Supplemental figures S2-5).

2. In figure 2C it is almost impossible to read some of the labels in the volcano plot.

Authors: The letters in the volcano plot (Fig. 2C) have now been enlarged (updated Figure 2 provided below)

3. The data presented in figure 4 show species level correlations between various metabolic and phenotypes. However, unless I am missing it, I do not see any details or description of the methods used for plasma glucose and insulin levels, or immunohistochemistry of Ki67 or at F4/80 staining in the colon tissues in the methods section. This should be included.

Authors: These phenotypes and markers were generated in our previous study (Hases et al, 2020). A summary of the methods has now been added to the Material and Methods, and the Figure 4 legend has been clarified and reference added (see below).

4. In the same figure 4, best fit or confidence intervals might benefit the reader in addition to the R and P values provided.

Authors: This has now been added (updated Figure 4 provided below).

5. On page 20, it is unclear if littermates were used in all studies.

Authors: This has now been clarified in Material and Methods p.21.

Reviewer #3: The presented studies were designed to determine if sex differences influences gastrointestinal microbiota populations and whether estrogenic ligands can attenuate high fat diet (HFD)-induced dysbiosis. Whole-genome shotgun sequencing (WGS) was used to define microbial composition of cecal samples in an effort to characterize the impact of HFD, gender, and estrogenic ligands and the interplay between the variables. Sex differences were observed at baseline and HFD induced changes in microbiota populations, both of which support previously published studies. Estrogen ligands had some effects particularly in male mice and some novel data is presented. However, there are questions and concerns related to the study design and data analysis that should be addressed to enhance the manuscript.

Major Concerns:

1. One primary concern is the lack of description for the methodology for how procedures were followed to minimize differences in the microbiota of test animals that were not relevant to the hypotheses proposed. The authors discuss how efforts were made to account for cage effects statistically. However, best practice is to employ methodology (i.e. mixing bedding between cages, co-housing prior to start of the study, autoclaving bedding and water, etc...) that minimizes such differences. What such methodology was used should be discussed.

Authors: The experiment was performed under specific pathogen-free (SPF) conditions in an animal facility with a highly controlled environment). This includes that personnel wears sterile clothes and air-shower before entering the barrier. Bedding and cages are sterilized by autoclaving, the diet is irradiated while the water is filtered through 4 different filters and sodium hypochlorite is added. The cages are individual ventilated cages (IVC), either the Tecniplast, GM500 (Type II long) or Allentown, European Type II long 12V cage. Work with animals is performed in a laminar flow (LAF) system to prevent contamination. We have now expanded the methodology description to reflect these precautions. However, mixing bedding between cages or co-housing prior to start of the study, was not applied and this can be viewed as a limitation. We have now added discussions of this limitation (p.19): *“A limitation of our study is that we did not take additional precautions to further limit differences in microbiota between cages, such as mixing bedding between cages or co-housing prior to the start of the study.”*

2. The biological relevance of the estrogen ligand exposure is also a concern. It is not clear what exposing an intact female or male mouse to estradiol models for. In both cases, other hormones will be elevated which is likely to confound the data and make it difficult to interpret. This would apply to a lesser extent to treating with DPN in the same animals. More discussion should be provided as to why this experimental design was used as opposed to using the ligands in a low estrogen environment (i.e. ovariectomized mice).

Authors: The rationale for our design was to enable dissection of sex differences (microbiome) and including in the response to HFD. Next, we were interested in the extent that estrogen contributed to potential sex differences. Also, high-fat diet has been reported to contribute to increased colorectal cancer rates among young adults (Loomans-Kropp, 2019), while estrogen (oral contraception) has been reported to reduce incidences in premenopausal women (Fernandez, 2001, and Amitay et al, 2022). Thus, to enable interpretation relating to premenopausal women, it is preferable to use intact female mice.

Using ovariectomized females would not provide the answer to our main research question (innate sex differences, impact in premenopausal women). In mice, ovariectomization actually results in E2 levels significantly lower than in males (approx. 40%, Saito et al, Circ Res. 2009).

While other hormones indeed can be elevated as a result of estrogen signaling, we found no significant alterations between intact females given estrogen compared to those given vehicle. Thus, the potential elevation of other hormones did not seem to affect the microbiome.

Our study layout did allow for a clear interpretation of the outcome in relation to our aims. The rationale has now been clarified also in the discussion. (p. 13) and a new section has been added discussing the impact of these findings (pp. 20-21).

3. Because estrogen is thought to influence it in rodents, was food intake measured? Changes in food intake might influence the microbial composition of the gut.

Authors: We agree with the reviewer that a changed diet consumption could impact the microbial composition. Therefore, a measure of food intake was included, which did not show differences in food intake between HFD and HFD+E2 (previously reported in Hases et al, 2020). However, the measurement was difficult because the high-fat content of the diet caused it to easily break into a powder at the temperature of the animal room (23 C). Consequently, “powdered diet” was found in the cage that we could not fully consider in our calculation, and we cannot exclude that this impacted the measurement. We have now addressed the consideration of food intake in the discussion (pp 19-20): *“Further, estrogen has been reported to impact the feeding behavior⁴², which*

in turn could impact the microbiome. However, as food intake was not successfully monitored in this study (as reported previously,¹⁷), we cannot exclude the possibility that some of the modest effects by estrogen could be through decreased diet consumption.”).

Minor concerns:

1. There are a few minor typographical errors that should be addressed in editing.

Authors: We have edited the manuscript throughout (but all minor grammatical edits are not highlighted).

Updated figures 1 and 4 on next pages

Figure 1: Sex-dependent responses to HFD. (A) Boxplots presenting the significance of the beta diversity between groups, with Bray Curtis distance. (B) Scatter plot of the microbial rank in males against the microbial rank in females upon HFD. The upper right corner indicates top-ranked species in both sexes. (C) Volcano plots show the significantly altered species by HFD in females and males, respectively. (D) Boxplots of the additive-log-ratio transformed taxa showing the significantly altered species by HFD (blue) in females and males. * FDR<0.05, ** FDR<0.01, *** FDR<0.01 (Benjamini-Hochberg-adjusted p-value).

Figure 4: Correlation between bacterial species and metabolic or phenotypic parameters.

Bacterial species that were differentially abundant between the sexes and impacted by ligand treatment in males showed a correlation with (A) plasma glucose levels (n=8 for each sex and diet condition), (B) insulin levels (n=7-8 for each sex and diet condition), (C) visceral adipose tissue (VAT) in relation to total fat (n=8 for each diet condition in males), (D) with Ki67 immunohistochemistry (IHC) proliferation score (n=8 for CD and n=6 for HFD, sexes combined), and (E) with F4/80+ macrophage colonic infiltration (n=6 for CD and n=5 for HFD, sexes combined). Metabolic and phenotypic parameters were previously analyzed and reported.¹⁷ (Pearson correlation coefficient, r, p-value from student t-test, best fit line was fitted with 95% confidence intervals).

REVIEWERS' COMMENTS:

Reviewer #1 (Remarks to the Author):

The paper is ready for publication in its current form.

Reviewer #2 (Remarks to the Author):

I have no further comments. The authors addressed all prior concerns. I congratulate the group on a nice study.

Reviewer #3 (Remarks to the Author):

The authors have sufficiently addressed the previous comments.